# Recycling of Hard Disk Drive Platters via Plastic Consolidation

**DOI:** 10.3390/ma16206745

**Published:** 2023-10-18

**Authors:** Tomasz Skrzekut, Maciej Wędrychowicz, Andrzej Piotrowicz

**Affiliations:** 1Faculty of Non-Ferrous Metals, AGH University of Science and Technology, A. Mickiewicza Av. 30, 30-059 Krakow, Poland; skrzekut@agh.edu.pl; 2Institute of Materials and Biomedical Engineering, Faculty of Mechanical Engineering, University of Zielona Gora, Prof. Z. Szafrana Street 4, 65-516 Zielona Gora, Poland; a.piotrowicz@iimb.uz.zgora.pl

**Keywords:** HDD platters, plastic consolidation, recycling, mechanical properties

## Abstract

The paper presents the comparison of two methods of recycling aluminum from HDD platters—the melting method and the method of plastic consolidation. The main elements of HDD memory, i.e., data carriers (platters), were examined via the percentage share of the total HDD mass and also via EDS analysis. The most common are platters made of the aluminum alloy series 5XXX, which are covered with a thin magnetic layer made of nickel. The research involved removing data carriers from about 30 HDDs and fragmenting them. The next step was to divide the platters into three groups; one was melted, the second was subjected to plastic consolidation, and the third group was fragmented into chips and also subjected to the consolidation process. Then, in the process of co-extrusion, rods were extruded from each material, and were subjected to EDS analysis, microstructure testing, Vickers hardness, and uniaxial tensile tests, and then the obtained results were compared. The obtained results of the microstructural tests in the case of gravity cast material confirmed the presence of the Al3Ni globular phase in the matrix. In the case of pressed and extruded materials, the Al3Ni phase appeared at the Ni-AlMg contact. After plastic consolidation, all the tested rods were characterized by their comparable strength properties (a tensile strength of 250 MPa and yield strength of 105 MPa).

## 1. Introduction

A hard disk drive (HDD) is the main and most popular storage medium for data and information in a computer, especially in laptops and any portable-type computers. The most important characteristics of the HDD are to store huge amounts of information (bytes, up to 22 TB and more [1]), it retains the saved data when the power supply is turned off, and it is able to read and write data countless times in a short period. The HDD is an extremely precise, durable, and advanced device [2].

The platter is one of the most important parts of HDD, because information is physically stored on it. It must be made of a lightweight yet rigid material (as opposed to the flexible materials used to make floppy disks), so aluminum alloys are the most suitable materials, although glass and glass composites have become popular in recent years [3]. The platter is covered with a magnetic layer on which magnetic impulses are recorded, which, thanks to the head, are read as single bits. The head and the surface of the platter are separated by a microscopic space, and the platter spins at a speed of between 5000 rpm and 7200 rpm [1]. These parameters mean that the surface of the platter must be perfectly flat and smooth, otherwise the surface of the platter or the head could be damaged, resulting in the loss of stored information. The method of applying the magnetic layer to the disk material is also important, as it translates into uniformity and surface roughness [4].

The magnetic layer of platters, on which the data is recorded, is applied to the surface most often via magnetron sputtering, vapor deposition or, less frequently, by surface galvanizing [5]. In the case of old-type HDDs, a coating consisting mainly of nickel is applied to the surface of the platters, but a protective layer is also applied, usually made of aluminum oxide, giving a mirror image, and a very thin layer of perfluoropolyether, which has the function of leveling the coefficient of friction during contact between the head and the platter [6]. Some HDDs have more than one platter, and each platter can be written on both sides, which proportionally increases the capacity of a single memory unit. However, as the number of platters increases, the energy required to spin them increases, so, currently, there is a maximum of three platters per disk unit. Platters larger than 3.5 inches in diameter are also not currently used as, with larger diameters, there is a problem with maintaining rigidity [1,2,7].

The table below shows the average weight and percentage share of individual raw materials (Table 1) in the total weight of the HDD, respectively. The predominant metal in HDDs is aluminum, as it makes up more than 50% of the total weight. This metal is used as base castings [7], to make platters (6.0–7.8%), and the chassis (46.8–51.9%), as well as the cover (11.6–21.6%) and the distance between the platters. 

The magnetic screens of magnets, magnets themselves, and sometimes top sheets are made of ferromagnetic materials. It is difficult to determine the exact composition of the ferromagnets present in the disk, as there are different kinds. We can find NdBFe or SaCo magnets and different contents of rare earth metals (for example dysprosium or samarium) [9,10,11,12]. Some disk closures are made of various types of metal sheets glued together, including stainless steel, aluminum, or ferromagnetic materials. The stainless steel content in the composition of the disk is about 10%, because it is used, among others, for all mounting screws and mounting plates on the motor. Table 1 also shows us the percentage of all electronics used in the HDD, and the weight of these components is approximately 8% of the total weight of the unit. This value is made up of various materials, from plastics to copper to small amounts of precious metals [7,8,9]. Drive components, including the motor, head arm, plastic elements, electronics, gaskets or moisture absorbers, and filters, account for almost 13% of the overall weight, but these are the parts that are very light and difficult to disassemble and have a diverse composition. An important feature of all raw materials obtained during dismantling is their high purity, i.e., they are high-quality raw materials.

The chemical composition of HDD platters may vary slightly depending on the manufacturer, the method of coating, or/and the weight of the platter. The mass share of the platter itself in the total mass of the HDD ranges from 3% to almost 5%, which mainly depends on the amount of media placed inside the drive. The main component of magnetic platters is, of course, aluminum, which accounts for about 95% of the platter’s weight [8,9,11,13].

Since the beginning of the 21st century, a competitor to HDDs has appeared on the commercial market—the SSD. A solid-state drive (SSD) is an all-electronic (it means flash storage and has no moving parts) non-volatile storage device that is an alternative to, and is increasingly replacing, hard disks. SSDs are faster than HDDs because there is zero latency (no read/write head to move). They are also more rugged and reliable and offer greater protection in hostile environments. In addition, SSDs use less power and are not affected by magnets [14]. Generally, SSDs have the same advantages over HDDs.

In time, there will only be solid-state storage, and spinning disk platters will be as obsolete as the punch card. Therefore, more and more HDDs are starting to appear in the new stream of electronic waste. When the term “HDD recycling” is used, it almost always means the recovery of neodymium as the most desirable component of HDDs [9]. Suffice it to say that the neodymium is a critical raw material for the European Union (EU) [15,16], and neodymium concentrate from the recycling of HDDs is richer in Nd content than the neodymium ore; hence, the recycling of neodymium is perceived positively not only from an ecological and economical, but also from a technological point of view [10,17,18].

In general, the known methods of HDD recycling can be divided into three categories: pyrometallurgical, hydrometallurgical, and decrepitation methods. Almost all of these methods require prior enrichment, such as dismantling, demagnetization, or/and mechanical separation via sieving [17] (although it is not a rule [19]). Depending on the stage at which the recycled HDDs come into the material flow, pyrometallurgical recycling can be based on any of the following strategies [17]: (1) material recycling, in which scrap materials are charged into smelting processes as raw materials; (2) alloy recycling, in which the materials are recycled into master alloys for magnet production; and (3) magnet recycling, in which magnet alloys are recycled in their current form. In hydrometallurgical methods, in principle, it is always about acid leaching, e.g., with sulfuric acid [19,20,21] or organic acids [22,23], followed by the precipitation of neodymium salts [23,24]. In [6], hydrochloric acid was used to dissolve the substrate of an HDD aluminum platter. Recycling via decrepitation involves the reaction of a neodymium magnet with hydrogen gas. The reaction of neodymium with hydrogen produces neodymium hydride, manifested by the disintegration of the magnet. The neodymium hydride will then separate mechanically. Neodymium hydride is a substrate for the production of new neodymium magnets [25,26].

A relatively new and niche method of recycling aluminum is plastic consolidation (PC). It is a process in which the use of high temperature is omitted during material processing. The process of PC occurs spontaneously when material particles are compressed with force until they come into such close contact that the short-range interaction is transformed into a permanent interatomic bond. The most effective example of PC is the extrusion of the material at an elevated temperature, where material particles are brought into contact under conditions of specific stresses and strains [27]. In other words, plastic consolidation is a set of technological operations during which dispersed metal forms are transformed into a solid material. Cohesive bonds between the material particles are formed and, as a result of PC, we can obtain a material that is characterized by a lower porosity and smaller grain size [28].

The aim of PC is to obtain a material that is very similar in properties and appearance to the cast material, but omitting the liquid phase, which limits diffusion processes and unfavorable grain growth in the material [29]. This method is most commonly used in powder metallurgy to densify the powdered material during the compaction process. The PC process can be carried out not only via traditional extrusion, but also via other deformation methods, such as rolling, forging, hydro-extrusion, and special methods of plastic working [30,31,32,33,34].

In the case of recycling aluminum, it should be remembered that the most effective method is the remelting of large parts of aluminum scrap. In the case of components such as chips, this is more problematic, due to the fact that after machining processes, chips are often contaminated with coolant and their surface oxidizes quickly, which causes a decrease in the efficiency of the melting process. Remelting the chip after the machining process, such as cutting, turning, or milling, is very unprofitable because, at the end of the process, we will have achieved only 40% of the input material mass, to obtain up to 100% of the mass of the finished product, compared to the mass of the input material [35,36].

PC allows for the minimization of energy consumption and also saves time and minimizes the number of technological processes needed to prepare and carry out the method compared to the smelting process. For small-batch or unit production, PC consists mainly of the initial compaction of small parts of the material, most often in a hydraulic press and, under the influence of a certain force, disks or briquettes are obtained, which then become part of the input material for the extrusion process. The pressure of the press, as well as the parameters of the subsequent extrusion, are conditioned according to the morphology of the crushed parts and the properties of the material. Prior to extrusion, the batch material is usually heated to a certain temperature, which is above the solvus line temperature, but must be below the liquidus line temperature. By heating to an elevated temperature and selecting the appropriate parameters of the extrusion process, we can obtain products with appearances and properties very similar to those of products made of cast solid material [27,28,37].

Many scientific papers on the recycling of HDDs focus exclusively on the recycling of rare earth elements from HDD permanent magnets (as presented and highlighted in [9], for example), due to the fact that rare earth elements are critical raw materials for the European Union and, therefore, their recycling is in high demand [4]. Another article [38] focused on the issue of recycling not only rare earth metals, but also copper, from HDDs. However, as highlighted in this article, aluminum makes up the majority of HDDs’ weight, so HDD scrap should be treated as a secondary source of aluminum.

The authors of this article are aware of articles dealing strictly with the recycling of aluminum from HDD platters in a hydrometallurgical way [6]. Each recycling method, regardless of the type of material processed, has its advantages and disadvantages, and it is difficult, with so many processing methods available, to clearly state that one method is the best, especially “in the form of digits”. The intention of the authors of this article was to investigate another, alternative method of recycling aluminum from HDD platters, using an unconventional plastic consolidation method. 

## 2. Materials and Methods

The subject of the study comprised HDD platters obtained from worn or non-functional HDDs (Figure 1A). The drives were manufactured by companies such as Samsung, Seagate, Toshiba, and Hitachi. Thirty HDDs were used for the tests, from which approx. 600 g of data carriers, i.e., platters alone, were obtained (Figure 1B). The platters were removed by hand. The disks contained one or several data carriers, depending on the capacity of the disk.

All the platters (600 g) were divided into three groups. The first group consisted of platters (about 200 g), which were cut into smaller elements using a guillotine (Figure 2A) and then inductively melted in a protective atmosphere of argon and gravity cast into molds with a diameter of 38 mm. After casting, the charge (Figure 3A) was homogenized at 550 °C for 12 h. The second group of platters (200 g) were cut with a guillotine into small pieces from 5 to 15 mm in length and 6 mm in width (Figure 2C), and the resulting material was divided into 6 portions of approx. 30 g each. Each of these parts of the material was pre-compacted on a KHEPS 100 Georg KIRSTEN D-54427 Kello hydraulic vertical press (Kell am See, Germany) to form a briquette with a diameter of 38 mm and a height of approx. 10 mm (Figure 3C). Six such briquettes constituted the input to the extrusion process. The third group of platters (200 g) was milled using an end mill without the use of coolant to form fine chips (Figure 2B). Then, briquettes were produced from the chips, as described earlier (Figure 3B). In this way, 3 charges with a diameter of 38 mm and a weight of about 180 g were obtained and were subjected to the coextrusion process. Extrusion was carried out at a temperature of 375 °C at an extrusion speed of 1 mm/s. After the extrusion process, three rods with a diameter of 8 mm were obtained. The tests of the chemical composition of the rods after the extrusion process were carried out on the Foundry-Master-Pro 2 device (Ueden, Germany).

Samples were taken from the extruded rods and metallographic specimens and were prepared to observe the microstructure. The samples were ground on abrasive papers with a gradation of 180–1000 (paper), and then polished with diamond pastes (DP-Suspension P by Struers) with a gradation of 9 and 3 µm. Finishing polishing was performed using silica colloidal suspension (OP-S by Struers). Grinding and polishing of the samples was performed on a RotoPol 11 device (manufactured by Struers, Copenhagen, Denmark). Microstructure observations were performed using a Hitachi SU-70 scanning electron microscope (Hitachi High-Technologies Corporation, Tokyo, Japan). EDS spectroscopy was used to study the chemical composition in microareas.

X-ray diffraction analysis (XRD) was performed at room temperature using a Rigaku MiniFlex II apparatus (Takyo, Japan). Diffraction reflections were obtained in the angular range of 2θ from 20° to 80°. The diffractometer was equipped with a Cu lamp that generated characteristic Kα radiation of 1.5418 Å.

The tensile test was carried out at ambient temperature in accordance with PN-EN ISO 6892-1:2020-05 [39] on a Zwick Roel Z050 testing machine (manufactured by ZwickRoell Group, Ulm, Germany). Cylindrical samples were cut from the beginning, middle, and end of each rod using a Secotom-10 cutter (manufactured by Struers, Copenhagen, Denmark). The tensile test was performed on 3 samples for each material. Samples with a diameter of 6 mm and a measuring base length of 30 mm were deformed at a speed of 8 × 10^−3^ s^−1^. On the basis of the obtained tensile curve, the mechanical properties were determined. The paper presents sample tensile curves for each material along with mechanical properties including standard deviation.

The compression test was carried out on the MTS 880 testing machine (MTS Systems Corporation, Eden Prairie, MN, USA) at ambient temperature and at elevated temperature. Samples with a diameter of 8 mm and a length of 11 mm were used for the compression test. The samples were cut directly from the extruded bar using the aforementioned cutter. The Vickers hardness measurement was carried out with a load of 19.61 N using a Shimadzu HMV-2 T microhardness tester (Shimadzu Corporation, Kyoto, Japan). For each material, 10 measurements were performed, and the average value of the measurement was presented, taking into account the standard deviation.

## 3. Results and Discussion

Figure 4 shows the results of the observation of the HDD platter microstructure (cross-section). Images of the microstructure for three thicknesses are presented: A—810 µm, B—1300 µm, C—1800 µm. It was found that, regardless of the thickness of the platter, the thickness of the magnetic (nickel) coating is 10 µm. A nickel coating is applied to both sides of the platter, which increases the surface area capable of recording data. The matrix of the disk is an Al-Mg alloy, which we confirmed via the analysis of the chemical composition of EDS (Figure 4 and Table 2). No nickel-rich intermetallic phase was found between the Ni layer and the Al platter core.

In the case of platters shattered into larger pieces (Figure 2A) and then cast into an ingot (Figure 3A) and extruded into a rod, the microstructure of this material is shown in Figure 5A. During induction melting, the reaction of the nickel coating with the disk matrix took place, as a result of which, we no longer observe nickel particles in the material, but the Al3Ni intermetallic phase (Figure 6A and Figure 7 and Table 3). Importantly, the morphology of Ni-rich particles is very favorable; the intermetallic phase is in the form of round, globular particles with an average diameter of 1.2 µm. These particles are evenly distributed over the observed surface. In the matrix of the rod, there is an Al-Mg alloy with a Mg content of 4% (Figure 6A and Table 3). The Al3Ni phase, next to such phases as Al-Cr, Al-Mo, Al-W, Al-Ti and others, is occurring in the so-called master alloys [40]. In an Al-Ni system, the temperature of 900 °C corresponds to the content of 17 at.% Ni (30 wt.% Ni). At that temperature, the Al3Ni phase is precipitating from the liquid; it is present as a constituent of Al + Al3Ni eutectic, crystallizing at a temperature of 650–640 °C [40,41,42], and this phase is present in the above-mentioned remelted material. In another article [43], research was carried out on the melting of aluminum and nickel and the formation of different phases during joining. As well as the Al3Ni phase, other Al-Ni phases have been disclosed, namely Al3Ni2, AlNi (stoichometric), AlNi (Ni-rich), and AlNi3. Due to the tested bonding method, the metals and their phases were arranged in layers and were not present in the entire material (they were not homogenized).

The microstructure of the rods after the plastic consolidation process is shown in Figure 5 and Figure 6. Depending on the method of recycling the output platters from HDD, we observe a diverse morphology of Ni particles in the Al-Mg matrix. The tests of the chemical composition of the rods after the extrusion process are presented in Table 3.

In the case of chips after milling (Figure 2C, Figure 5B, and Figure 6B), nickel particles with sharp edges are observed in the material and are unevenly distributed. The thickness of the Ni particles is comparable to the thickness of the nickel coating on HDD platters. The analysis of the chemical composition (Figure 6 and Table 4) confirms the presence of Ni in the Al-Mg alloy matrix (Figure 6B and Table 4). At the Ni-AlMg matrix boundary, an Al3Ni intermetallic phase was formed (Figure 6B and Figure 7 and Table 4). It is worth noting that it was not observed on the cross-section of hard disk platters (Figure 4), which proves that it was formed only during plastic consolidation in the extrusion process. The thickness of the Al3Ni phase around Ni was 0.45 µm on average. Figure 5C shows the microstructure of a rod extruded from HDD platters that had previously been milled to very fine chips (Figure 2B). During the extrusion of platters cut into pieces with a length of 5–15 mm and a width of up to 6 mm, the outer layers of the cut disks connected to each other during consolidation. Therefore, the nickel particles have a thickness of about 20 µm, and the Al3Ni intermetallic phase was also formed on their contact with the Al-Mg matrix (Figure 5C, Figure 6C, and Figure 7 and Table 4). This phase has an average thickness of 0.75 µm. Some of the Ni coatings crack, while, after extrusion, the Al3Ni phase forms in the place of cracks). This microstructure is comparable to the one described earlier; however, several differences can be seen. Firstly, in most cases, nickel particles consist of two layers (Figure 5C and Figure 6C); they do not occur singularly in the matrix, as was the case with the previously described material (Figure 5B).

Figure 8 shows exemplary stress–strain curves for samples made of the tested materials. The highest tensile strength is characteristic of the material melted and gravity cast and then extruded. It reached a tensile strength of 271 MPa (Figure 8 and Table 5). The elongation for this material is over 20%. Such good strength properties are the result of the even distribution of particles of the Al3Ni phase and its homogeneity (Figure 5A). The lowest value of tensile strength (237 MPa) is characteristic of a rod extruded from HDD platters cut on a guillotine and pressed (Figure 2C, Figure 3C, and Figure 8 and Table 5). It is worth noting that both the pressed and extruded materials have a similar elongation value (over 30%). The material obtained from chips after milling had the highest yield strength; the value of the yield strength in this case was 116 MPa (Figure 8, curve B). Curve A (Figure 8) is characterized by the presence of a yield strength of 95 MPa (Figure 8 and Table 5). The stress–strain curves look quite different in this respect for the other two materials (curves B and C). The pressed and extruded materials are characterized by the presence of a sharp yield point (Figure 8 and Table 5). This phenomenon may be caused by the presence of sharp edges of Ni-rich phases, which may propagate microcracks and reduce strength properties.

All tensile curves are characterized by the presence of a step change in stress in the material subjected to tension—the Portevin–Le Chatelier effect (PLC effect), which is typical of the 5xxx series [44,45,46]. There are many theories explaining this phenomenon [46,47,48]. The literature distinguishes three types of oscillations occurring on the stress–strain curve [48]. On all the stretching curves (Figure 8), the appearance of the PLC effect shows a similar course. We are dealing with a mix of oscillation types. In the initial phase, type B oscillations occur (cyclical but irregular), while, after exceeding the tensile value, for A-140 MPa, for B-114 MPa, and for C-105 MPa, the oscillations are characterized by sudden and irregular jumps, so type B changes to type A. Mixed types of oscillations during one tensile test can also be found in the literature [49,50]. The mechanical properties of the bars after the consolidation process are lower compared to the gravity cast material; however, they meet the requirements of the PN-EN 485-2+A1:2018-12 standard [51]. This proves that the plastic consolidation parameters were well selected, which is also confirmed by the density results presented in Table 5. The higher elongation in the case of plastically consolidated materials (curves B and C) compared to the gravity cast material (curve A) is caused by the presence of large areas of the plastic matrix, which can be seen in Figure 5. Similar research results can be found in the literature [36,52].

In the literature, we can find articles on the plastic consolidation of fragmented metal forms, in which the fragmentation of the charge material via mechanical processing allowed the obtention of similar- or higher-strength properties compared to solid material. In the work [52] in which the AlSi 11 alloy was tested, plastic consolidation allowed to improve the tensile strength by 50 MPa compared to the solid material. In the papers [28,53], the authors examined aluminum alloys 6082 and 1370. Plastic consolidation allowed the obtention of strength properties comparable to those of a solid material, as in the analyzed article.

Tests of the surface topography of material fractures after the tensile test were carried out, mainly to check whether the extrusion process with selected parameters was an effective way to obtain a material with good cohesion (Figure 9). In the case of the samples cast by gravity and extruded from the HDD platters (Figure 9A), the materials after stretching show the characteristics of a plastic fracture, with characteristic depressions in the middle in which there are fine particles of the intermetallic phase Al3Ni with a size of 1–2 µm. Cracking is initiated at the boundary of the separation of individual cavities. On the other hand, in the case of the crushed, pressed, and extruded HDD platters (Figure 9B,C), much larger cavities (approx. 100 µm) are visible on the surface of the observed fracture, inside which there are cracked Ni particles of varying structure in terms of both size and shape. Most Ni particles have sharp edges.

Examples of the influence of elevated temperature on the mechanical properties of the tested materials are shown in Figure 10. The mechanical properties of the extruded samples were tested in the temperature range of 20–500 °C by means of a compression test. It can be concluded that the type, size, and structure of the particles (Al3Ni in the case of gravity cast and extruded materials and Ni in other cases) do not significantly affect the change in strength parameters, and that these materials show a significant plastic deformation capacity at a high temperature. Figure 10D shows the yield strengths depending on the temperature at which the high-temperature compression test was carried out. On the basis of the presented tests, it can be concluded that in the temperature range of 25–300 °C, the highest YS value is observed for the material cast by gravity and extruded. In the case when the HDD platters were crushed, pressed, and extruded, the YS value is comparable for both materials across practically the entire temperature range.

Traditional processes for recycling aluminum, magnesium, and their alloys involve inevitable metal losses and require significant primary energy inputs, although these are an order of magnitude lower than those in the production of primary metal [27,37,54]. In the case of thick scrap, i.e., scrap with a high volume-to-surface ratio, approximately 2.0–2.6 kWh of energy is needed to recycle 1 kg of aluminum via melting. Energy is used almost exclusively for furnace and heating processes. The losses occurring in such a process range from 15% to 25% of the mass of the metal charge, including losses occurring during preparatory operations for the process. The results of recycling shredded scrap (cutting chips, thin wires, foils) via melting are much more unfavorable. Material losses in a traditional smelting process can be up to twice as high and, in some cases, only about 40% of the mass of the metal charge can be recovered. Such a low efficiency of shredded scrap processing results from the high chemical activity of aluminum at high temperatures when heated to melting. This makes it practically impossible to recycle very small metal fractions, which go almost entirely to the dross, constituting a burdensome waste of the process [27,55]. Theoretically, it is possible to extrude profiles without the need for additional preliminary heat treatment. In this case, the minimum energy needed for the extrusion process consists of the energy needed to preheat the charge and the energy needed to deform the material through the die. In the case of recycling using the extrusion method, 1 kg of aluminum requires at least 0.44 kWh of energy. However, the value of this energy depends on many factors, including the type of extruded alloy, the shape of the product after extrusion, and the force needed for extrusion.

## 4. Conclusions

The proposed recycling method makes it possible to use used HDD platters to produce, by means of plastic consolidation in the process of extrusion, a composite based on AlMg, whose reinforcement is made of Ni-rich particles.

The remelting of HDD platters and extrusion results in the obtention of a composite reinforced with very fine particles of the intermetallic phase Al3Ni, while the fragmentation of the platters and then pressing and extrusion lead to the obtention of reinforcement in the form of Ni particles and, at the boundary between the AlMg matrix and Ni reinforcement, the Al3Ni intermetallic phase is formed.

Composites obtained via the low-energy method (without the melting process) are characterized by comparable strength properties, and the type, structure, and size of the Ni particles do not affect the strength properties tested at elevated temperature.

## Figures and Tables

**Figure 1 materials-16-06745-f001:**
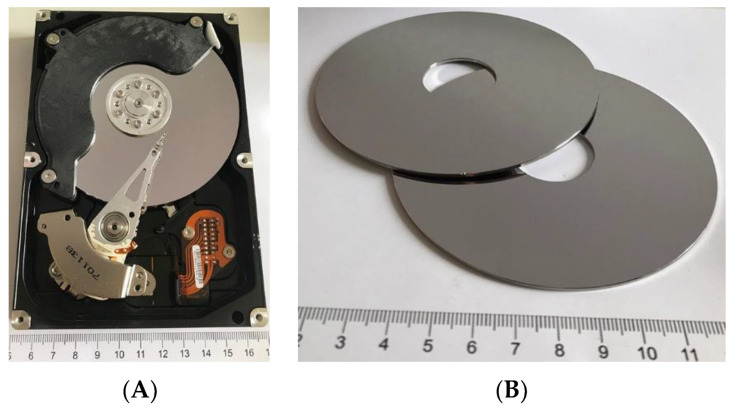
HDD: (**A**) in its initial state, (**B**) with platters removed.

**Figure 2 materials-16-06745-f002:**
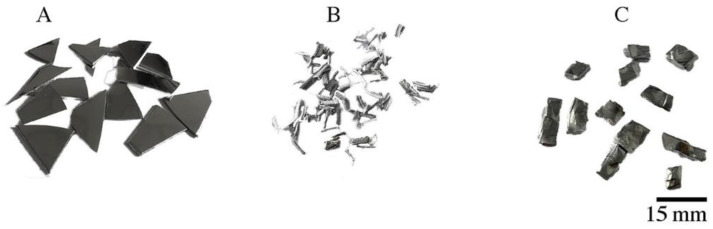
Platters after fragmentation with the help of: (**A**,**C**) a guillotine, (**B**) a milling cutter.

**Figure 3 materials-16-06745-f003:**
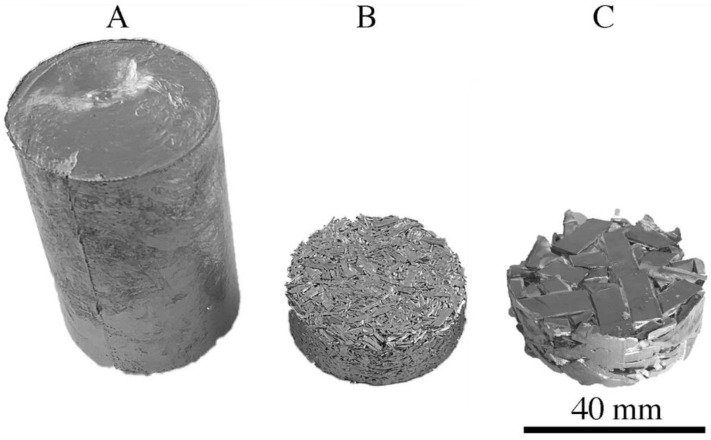
Charge before extrusion: (**A**) after gravity casting, (**B**,**C**) after pressing.

**Figure 4 materials-16-06745-f004:**
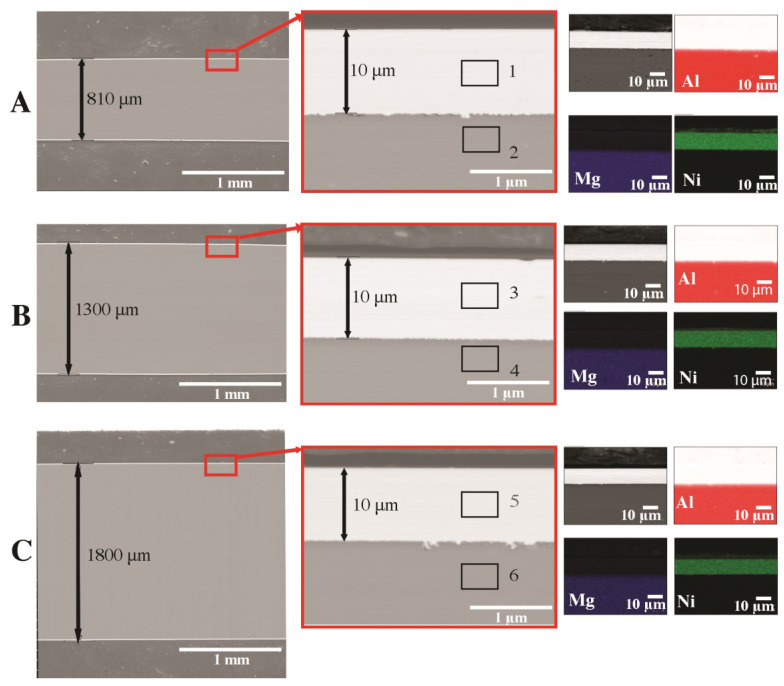
The microstructure of HDD platters (cross-section): SEM with the EDS chemical mapping of components: (**A**) plate with a thickness of 810 µm, (**B**) plate with a thickness of 1300 µm, (**C**) plate with a thickness of 1800 µm.

**Figure 5 materials-16-06745-f005:**
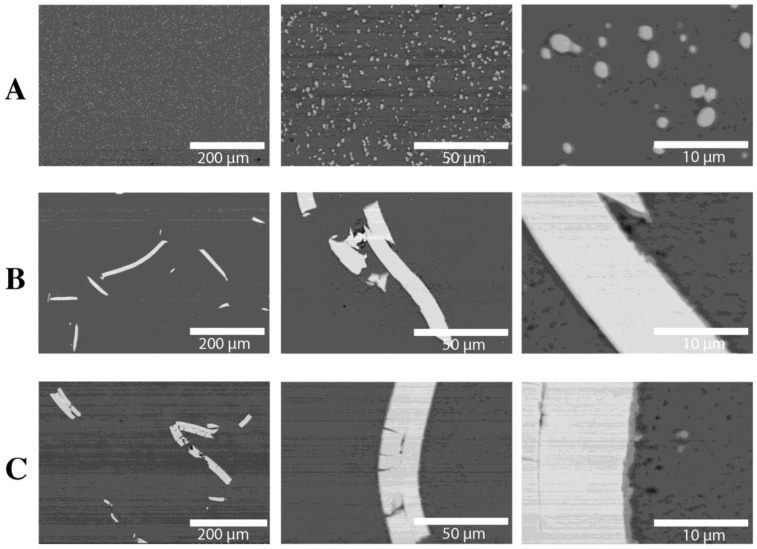
Microstructures of extruded rods: (**A**) HDD platters cut via guillotine, melted, and gravity cast, (**B**) HDD platters milled and pressed, (**C**) HDD platters cut via guillotine and pressed.

**Figure 6 materials-16-06745-f006:**
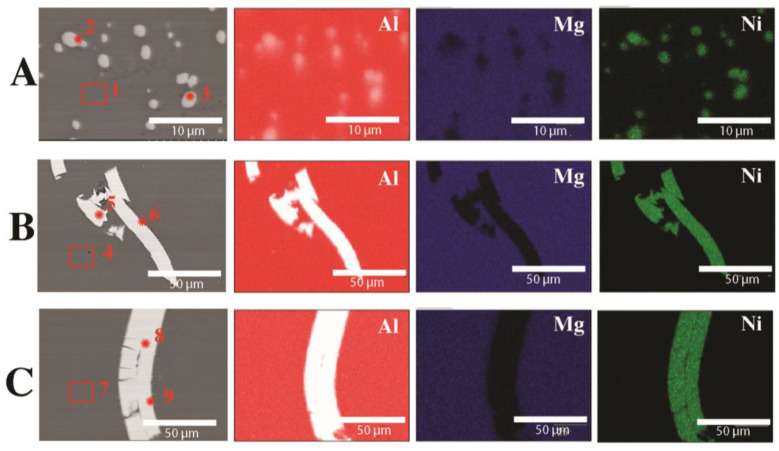
Analysis of the chemical composition of extruded rods (SEM): (**A**) HDD platters cut via guillotine, melted, and gravity cast, (**B**) HDD platters milled and pressed, (**C**) HDD platters cut via guillotine and pressed.

**Figure 7 materials-16-06745-f007:**
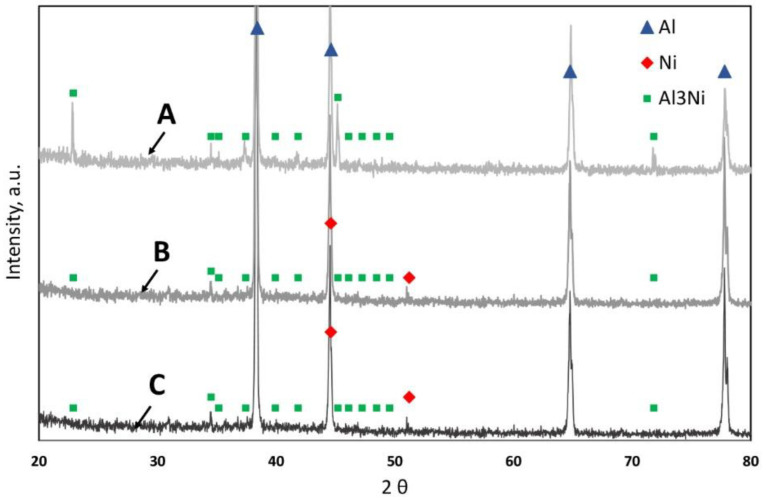
The X-ray diffraction pattern of rods after the extrusion process: A—HDD platters cut via guillotine, melted, and gravity cast, B—HDD platters milled and pressed, C—HDD platters cut via guillotine and pressed (ICDD: Al 00-004-0787; Ni 00-001-1260; Al3Ni 00-002-0416).

**Figure 8 materials-16-06745-f008:**
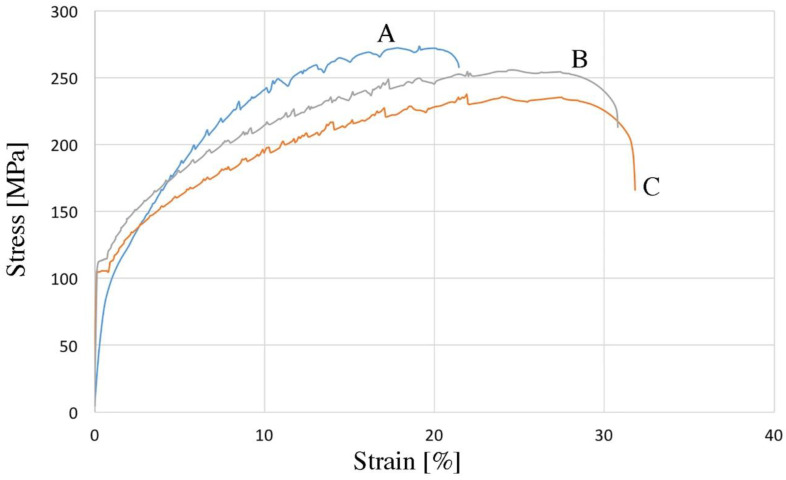
The stress–strain curves of samples after the extrusion process: A—HDD platters cut via guillotine, melted, and gravity cast, B—HDD platters milled and pressed, C—HDD platters cut via guillotine and pressed.

**Figure 9 materials-16-06745-f009:**
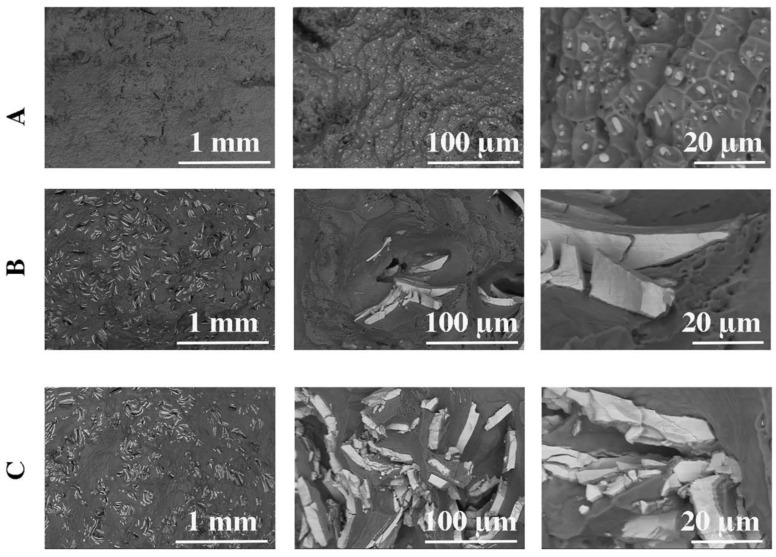
Fractures after the tensile test: (**A**) HDD platters cut via guillotine, melted, and gravity cast, (**B**) HDD platters cut via guillotine and pressed, (**C**) HDD platters milled and pressed.

**Figure 10 materials-16-06745-f010:**
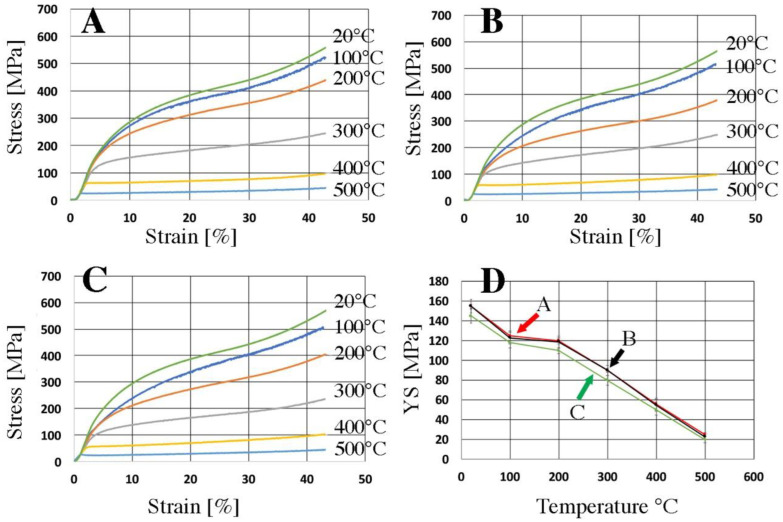
The stress–strain curves received for the as-extruded materials deformed via compression at a constant true strain rate. (**A**) HDD platters cut via guillotine, melted, and gravity cast, (**B**) HDD platters milled and pressed, (**C**) HDD platters cut via guillotine and pressed, (**D**) the yield strength depending on the temperature for the tested materials.

**Table 1 materials-16-06745-t001:** The average mass (g) and percentage (%) of HDD raw materials [8].

Raw Material	The Average Mass (g) and Percentage (%)	
Aluminum	264	51.3
Ferromagnetic materials	90	17.5
Stainless steel	53	10.4
Electronics	41	7.9
Sub-assemblies and others	66	12.9
Total	515	100

**Table 2 materials-16-06745-t002:** Results of X-ray chemical analysis (EDS) for 1–6 areas marked in Figure 4 (% by weight).

No	Mg	Al	Ni	Fe
1	0.0	0.1	99.9	0.0
2	4.1	95.8	0.0	0.1
3	0.0	0.1	99.9	0.0
4	4.2	95.7	0.0	0.1
5	0.0	0.1	99.9	0.0
6	4.1	95.7	0.0	0.1

**Table 3 materials-16-06745-t003:** The chemical composition of rods after the extrusion process: A—HDD platters cut by guillotine, melted, and gravity cast, B—HDD platters milled and pressed, C—HDD platters cut by guillotine and pressed.

Element	Al	Ni	Fe	Mg	Other
**A**	91.49	4.53	0.08	3.79	0.11
**B**	91.38	4.56	0.08	3.88	0.10
**C**	91.32	4.57	0.09	3.90	0.12

**Table 4 materials-16-06745-t004:** The results of X-ray chemical analysis (EDS) for areas 1–9 as marked in Figure 6, weight %.

Element	Mg	Al	Ni	Fe
1	3.8	95.9	0.3	0.1
2	0.2	61.4	38.4	0.0
3	0.3	61.5	38.2	0.0
4	3.9	95.8	0.3	0.1
5	0.0	0.2	99.8	0.0
6	0.1	58.9	41	0.0
7	3.8	95.9	0.3	0.1
8	0.0	0.1	99.9	0.0
9	0.1	63.4	36.5	0.0

**Table 5 materials-16-06745-t005:** The mechanical properties of the researched materials. Standard deviation is marked in italics.

Element	UTS, MPa		YS, MPa		Sharp Yield Point				Elongation, %		Hardness, HV2		Density g/cm^3^
					LYP, MPa		UYP, MPa						
A	271	±4.2	95	±4.1	-	-	-	-	20	±3.2	74	±2.2	2.76
B	254	±3.1	-	-	112	±3.3	116	±3.4	31	±3.0	72	±2.1	2.75
C	237	±2.9	-	-	104	±2.4	106	±3.0	32	±2.7	71	±1.9	2.75

## Data Availability

Not applicable.

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
