# Peer review of "Recycling of Hard Disk Drive Platters via Plastic Consolidation"

_materials, 2023, doi:10.3390/ma16206745_

Round 1

Reviewer 1 Report

The present manuscript deals with recycling of Al in HDD platters with an alternative method based on plastic consolidation. The work sounds as a technical report of experimental tests related to some commercial contract. The scientific quality of the results is far to be adequate for the publication on Materials and the presentation of the results demands a deep revision. Moreover, the topic explored by the authors could attract the interest of readers of other MPDI journals like Metals.

The most critical points are here listed:

1) The introduction is verbose providing unnecessary details on the technology of HDD and existing recycling methods. Moreover, the authors did not elucidate the reason why an alternative method should be necessary; the research of an unconventional way to recycle such special waste must be justified by valid assumptions.

2) HDD acronym is never explained so the reader not expert does not understand the origin of the aluminium-based platforms

3) The chemical analyses reported in tables 2 and 4 indicate the relatives amounts of Al, Mg and Ni but no e.d.s are included. It is recoomended to include the standard deviations of the quantitative analyses.

4) In the XRD shown in figure7 the symbols are labelling diffraction peaks according to solid state phases. The ideal positions, identified by diamond symbols, associated to Al3Ni phase vary in the three XRD presented. This is conceptually uncorrect. It is recommended to provide the ICDD reference for the calculated 2theta positions for the three Al, Ni, and Al3Ni phases. 

In general, this manuscript requires a massive revision and the authors should provide more strong evidences indicating that the method described is an alternative and valuable route for the Al recycling from HDD.

The English used is basically correct but in some parts of the text is lacking of professional exposure. 

Author Response

Dear reviewer,

The answers are in the attachment.

Reviewer 2 Report

It seems, that the idea of utilization of aluminium by suggested method with the use of compacting more or less the question of the total process cost, including the preparation stage, and quality of the product during technological process and its stability. here I can not find the data about reproducibility of the mechanical properties. Does the indicated values corresponds to several samples or only one for each series? What will be the total error of obtained values?

It seems strange that the elongation of the samples of serie B and C is much higher than for serie A. Can it be explained?

in the title of Figure 9 (line 348-349) there is a misprint (two times indicated serie B)

Author Response

Dear reviewer,

The answers are in the attachment

Reviewer 3 Report

The paper is devoted to the subject of energy-saving aluminum recycling of HDD parts, speciafically HDD platers.

It is undoubted that HDD recycling will become more and more important problem with larger spread of SSD. However, the Aluminum content in HDD platers seems marginally low to consider it a major problem. As authors state in table 1, there are 264 grams of aluminum in a HDD of which only 6-7.8% corresponds to platers. This comes to around 20g of aluminum per drive, so 1 thousand recycled HDDs would produce only 20 kilos of aluminum which is a negligible number in respect to the worldwide volume of scrap of aluminum parts measured probably by thousands of tons.

Second, aluminum is a low melting point material and thus would require much less energy for remelting compared to recycling of steels or other high melting point metals. Thus, it would be much more critical to develop energy saving technologies for these materials instead of aluminum. The manuscript provides no exact number or even estimate of how much energy is saved if the authors method is used compared to the traditional melting. 

Finally, the resultant properties after extrusion are even lower than those of remelt and cast parts. The microstructure of extruded scrap with coarse and sharp particles of the second brittle phase is very detrimental to properties in fatigue of fracture toughness. There is also no data on the porosity of the material. Much better properties of the material would have been achieved if cast part is then extruded and thus more energy spending would be meaningful to achieve higher strength. However, this topic has already been researched quite intensively.

All these points make the reviewer doubtful of the scientific importance of this manuscript. And the results are not outstanding to say the least. However, the paper is written in clear language and certanly presents some new findings on the stated subject. 

Final decision: There is no strong rejection or acceptance recommendation and the final decision should be made by the editorial board after consideration of other reviews.

If the paper is decided to be published, there are few technical moments to be addressed:

Lines 12 and 18: “chemical composition (SEM analysis)” is probably meant EDS analysis.

Line 63: “Magnetic screens of magnets, magnets and sometimes…” is recommended to be corrected as “Magnetic screens of magnets, magnets themselves, and sometimes…” for improved clarity.

Line 67: “Some disc closures are made of various types of metal sheets glued together, made of stainless steel, aluminum or ferromagnetic materials.” is recommended to be corrected as “Some disc closures are made of various types of metal sheets glued together, including stainless steel, aluminum, or ferromagnetic materials.”

Line 68: “Stainless steel in the composition” is recommended to be corrected as “Stainless steel content in the composition”.

Line 95: please provide a full transcription of the abbreviation “EU”.

Line 315: please provide a full transcription of the abbreviation "PLC".

Author Response

Dear Reviewer,

The answers are in the attachment.

Round 2

Reviewer 1 Report

N/A

N/A